# Social-Media Analysis for Disaster Prevention: Forest Fire in Artenara and Valleseco, Canary Islands

**Gorka Zamarreño-Aramendia** [1] , **F. J. Cristòfol** [2,*] , **Jordi de-San-Eugenio-Vela** [3] **and Xavier Ginesta** [3]

1   Department of Theory and Economic History, University Malaga, 29013 Malaga, Spain; gzama@uma.es
2   Department of Market Research and Quantitative Methods, ESIC, Business and Marketing School, 28223 Pozuelo de Alarcón (Madrid), Spain
3   Department of Communication, University of Vic, 08500 Vic, Spain; jordi.saneugenio@uvic.cat (J.d.-S.-E.-V.); xavier.ginesta@uvic.cat (X.G.)
*   Correspondence: fjcristofol@esic.edu; Tel.: +34-606-83-77-78

**Abstract:** This manuscript investigates the use of social media, specifically Twitter, during the forest fires in Artenara and Valleseco, Canary Islands, Spain, during summer 2019. The used methodology was big-data analysis through the Union Metrics and Twlets tools, as well as content analysis of posts related to the fires written by seven relevant accounts on the days when the fires were active, which was between 17 August and 26 September, when 9636.40 hectares were burned. The accounts selected for analysis were the following: Ángel Víctor Torres, autonomous president; Canary Islands Government; Civil Protection of Las Palmas; Military Emergency Unit of the Spanish Army; Delegation of the Spanish Government in the Canary Islands; Citizen's Service of the Canary Islands Government; and the information account of the Security and Emergency area of the Canary Islands Government. The study concludes that the Canary Islands authorities did not use social media as a preventive element, but almost exclusively as a live-information channel. Future recommendations are presented for the management of social media during natural disasters.

**Keywords:** natural disasters; big data; social media; Canary Island; forest fire

---

## 1. Introduction

Natural disasters are phenomena that exceed the expectations and habitual capacities of human beings, and have important consequences in the natural environment, in addition to requiring the management of large resources to confront them [1].

Catastrophic events are nothing new in the history of humanity. From the eruption of Vesuvius [2] that swept the cities of Pompeii and Herculaneum in 79, and the subsequent event of 1631 [3], to the earthquake and tidal wave that affected Lisbon on 1 November 1755 [4], and the earthquake and subsequent fire of San Francisco that destroyed the city [5] on 18 April 1906. In addition to these, there were the fire of Yellowstone Park in 1988 that destroyed 321,300 hectares [6], a seaquake that killed over 200,000 people in Southeast Asia in 2004 [7], the destructive 2010 seism in Haiti [8], and the recent fires that ravaged Australia and destroyed over 1 million hectares of land [9].

Climate change, which affects the entire planet, has highlighted the virulence of increasingly extreme weather events that cause natural disasters, and increase in frequency and intensity every year [10–13]. The consequences on millions of people and their effects on the environment are a global phenomenon [14,15], of which the results are difficult to evaluate and which spread to areas such as migration or political conflicts [16]. Thus, the interest in investigating the relationship between climate-change events and the use of social media has been increasing in academic fields [17–19].

The ability to respond to these catastrophic events is very different around the world, and there is a significant gap between developed countries and those at different stages of development, which translates into a direct effect on the number of affected people, fatalities, and the resources used for reconstruction and restitution of the affected areas [12].

The objectives of this research are as follows. First, analyzing how Canarian authorities use social media, specifically Twitter, in terms of natural risk prevention, and second, analyzing the management of information arising from social media on Canarian authorities' Twitter accounts during the fires of August 2019 (in the municipalities of Valleseco and Artenara). The following research questions (RQ) resulted from the objectives presented before:

RQ1. Do Canarian authorities use messages in order to prevent future natural disasters?
RQ2. Which digital-communication strategy have the Canarian authorities establish to manage the disaster?

After this introduction, this article presents a theoretical background to contextualize the significance of big data and social media for crisis management, especially regarding natural disasters (Section 2.1). Following this, Sections 2.2 and 2.3 contextualize the Canarian islands and the significance of the tourism industry for their economy. Section 3 descries the methodological framework of this research in order to analyze the impact of Twitter on communication management during the natural crisis of August 2019. Results, discussion, and conclusions are presented in Section 4, Section 5, and Section 6, respectively.

## 2. Theoretical Background

Communication is vital in times of crisis, with natural disasters being one of the fields where they play a major role [20,21], especially social media [22]. These communication channels play a complementary role to traditional media by adding the feature of immediacy, until now only offered by the radio [23,24]. They are also particularly relevant in terms of the information offered by government channels [25], as these institutions have realized the potential of these communication channels in order to address crises [26,27].

Social media play an increasingly important role in the organization of monitoring and coordination devices [28], as they did in the catastrophic floods in the states of Uttarakhand and Himachal Pradesh (India) [29], or in the Shouguang flood (China) [30]. The key role of social channels in monitoring activities may be further enhanced by opportunities in terms of data visualization and learning, which are aimed at preventing and/or improving overall management practices [31,32]. In addition, social media have interesting advantages over traditional media, as they provide a chance for the early detection of natural disasters [33], the coordination of relief and rescue actions [34], or obtaining real-time event information [35].

Social media offer a form of communication not only within the affected areas, but also between them and the rest of the world. They provide platforms for the rapid detection of natural disasters [33] and quick identification of the problem [36], which serve to properly manage help resources [34]. However, understanding the dynamics of social media with respect to the audience and the use of the information is a challenge that researchers are addressing from the aspects of big data and computer science [26,37–41].

### 2.1. Natural Disasters, Big Data, and Social Media

According to Martínez–Álvarez and Morales–Esteban [42], natural disasters' "prediction and characterization have been addressed from many different points of view. Most methods reported in the literature so far are based on statistical analyses of diverse geological indicators and certain precursory patterns". Restrepo-Estrada et al. [43] affirmed that, "Georeferenced social media messages are increasingly being regarded as an alternative source of information for coping with flood risks.

However, existing studies have mostly concentrated on the links between geosocial media activity and flooded areas". This georeference is provided by social-media platforms like Twitter.

As Wang and Ye [44] indicated, social media are a basic element in the management of natural disasters. There are four aspects to consider regarding social media and their involvement with natural disasters: time, space, content, and the network itself. According to the authors, most studies "involve multiple dimensions of social media data in their analyses", but there are also analyses separated by dimensions, as well as synchronous analyses for different dimensions. Lastly, "there are fewer simultaneous analyses as dimensions increase".

In this regard, Ghani et al. [40] stated, "big-data analytics has recently emerged as an important research area due to the popularity of the Internet and the advent of the Web 2.0 technologies". These authors express the importance of big-data analysis in the case of some natural disasters in order to create awareness, and its usefulness for observing and analyzing human behavior, specifically in the development of Hurricane Sandy. According to Pourebrahim, Sultana, Edwards, Gochanour, and Mohanty [45], "social-media platforms such as Twitter allow public and officials to share texts and photos, which can be a powerful means of communications during disasters". Kim, Bae, and Hastak [46] also agree on the importance of social media as a way of easier exchange of local knowledge between authorities and those affected by natural disasters. Lastly, Ukkusuri, Zhan, Sadri, and Ye [47] showed that social media are crucial for authorities and emergency management in order to inform and be informed of the reality on the ground.

Numerous studies link the relevance of Twitter to information before, during, and after natural disasters, and their possible prediction [20,21,48,49]. In cases of emergency, the appearance of influencers is relevant for the dissemination of information, according to Yang et al. [50]. In natural emergencies, it is necessary to identify users who publish objective information related to disasters in clear language and in a consistent manner.

Hernandez-Suarez et al. [51] presented Twitter as a "social sensor" in their study of the 2017 Mexico earthquake. Their work was based on the extraction of data during and after the aforementioned natural disaster. Ye et al. [52] conducted a multidimensional analysis of El Niño on Twitter: "Whenever Twitter users perceived what they thought were abnormal weather conditions, they immediately expressed their feelings and opinions on Twitter".

Kemavuthanon and Uchida [53], in their paper "Integrated question-answering system for natural-disaster domains based on social-media messages posted at the time of disaster", concluded that there is a possibility of creating a support system for foreigners residing in Japan to assist them in obtaining necessary real-time information during disasters in the future.

While there is a general understanding of the link between social networks and innovation activities with external stakeholders, research should be done on how and in what context social media can be used for open innovation throughout the innovation funnel. Howe [54] describes open innovation as "everyday people using their spare cycles to create content, solve problems, even do corporate R&D". Open innovation has also been described by Cachia et al. [55], differentiating and identifying three aspects of use in social media: creativity, expertise, and collective intelligence.

First, "creativity emerges from network interactions across of a mass of users with diverse knowledge (e.g., firms, consumers, universities, and any other social entity)"; second, expertise "refers to the ability of social media to provide an improved mechanism for insight and market foresight"; last but not least, collective intelligence "refers to the knowledge synergies that emerge from crowd collaborations on social media". However, Mount and Garcia Martinez [56] assumed that the majority of previous studies regarding the relationship between social media and open innovation "tend to focus only on ideation processes of utilizing social media for open innovation, such as idea competitions". In this research, we focus on the relationship among three elements: environmental risk management, social media, and open innovation at public administrations.

Open innovation processes always require a change in organizational culture. In this area of study, the academic contributions of Yun [57–59] are specially relevant. In particular, the paper of

Yun et al. [57] regarding the conceptual model of "culture for open innovation dynamics" is very interesting. The culture of open innovation dynamics presents the economy as a complex asynchronous and very dynamic system, so it is continuously developing in the micro- and macrodynamics of open innovation.

In the same way, and as is pointed out by Yun [58], open innovation must be controlled well by expanding the culture inside the organizations through a powerful leadership, and the institutionalization of this culture of open innovation.

### 2.2. Geographical Context

As Figure 1 shows, the Canary Islands belong to a group of Atlantic archipelagos along with the Azores, Madeira, Cape Verde, and the Savage Islands, known as the Macaronesian region. The Canarian archipelago is in front of the African coast, opposite Morocco and the Western Sahara, and seven islands compose it: Tenerife, Gran Canaria, La Palma, Fuerteventura, Lanzarote, La Gomera, and El Hierro. The archipelago also comprises seven islets, five north of Lanzarote, called Chinijo. All of them have a very similar morphology due to volcanic activity.

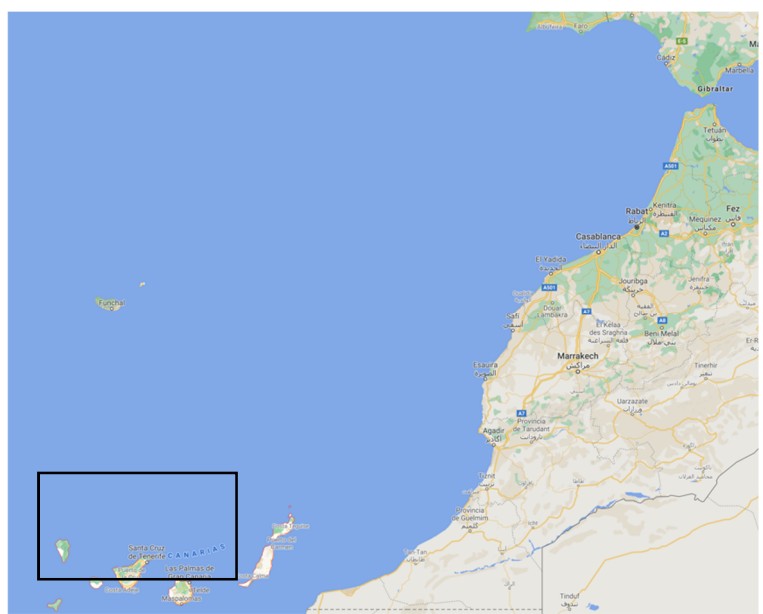

**Figure 1.** Geographical location of Canary Islands. Source: Google Maps. Edited by authors.

The Canary Islands are a chain of islands of volcanic origin that occupy a maritime area of about 100,000 km$^2$. The total area of the archipelago is 7446 km$^2$ (National Geographic Institute, 2020). The modeling resulting from volcanic activity gives them their significant altitudes (except for Lanzarote and Fuerteventura), with dominant central peaks and steep slopes. Teide, located on the island of Tenerife, is the highest mountain in all of Spain with 3718 m, while Gran Canarias' highest peak is Pico de las Nieves, with 1919 m.

The volcanic origin, the peculiarities of the relief, and the different climatic conditions that are present in each island create rich biodiversity. The uniqueness of their ecosystems, and their endemic flora and fauna, has meant having a network of protected areas, such as national parks (Garajonay, Las Cañadas del Teide, Timanfaya, and La Caldera de Taburiente) and a natural reserve divided among La Palma, El Hierro, Lanzarote, and part of the territory of Gran Canaria. Because of this, the Garajonay and Teide volcanos are part of a World Heritage Site [60].

The current relief of Gran Canaria is very diverse, resulting from its climatic evolution and its volcanic geology. There are different forms of relief, the first of which are massifs, which are very distinguishable territorial units, as they reach the coast in the form of cliffs coming from inland and

at heights of over 1000 m. These large massifs are Tamadaba, Altavista, Tirma, Inagua, Pajonales, and Southwest (Güigüí, Tasarte, and Tasártico). On the other hand, there are so-called volcanic mesas, which are the result of the erosion of ancient volcanic flows. This relief is above sea level and it is completed with volcanic cones resulting from volcanic eruptions, escarpments, and cliffs.

Other peculiar forms of the relief of Gran Canaria are ravines that are the result of erosion, and craters distributed throughout the island and give it its characteristic form. In the lower areas of the island, especially in the southeast, there are neutral relief forms with a gentle topography and relative extension, differentiating between plains, wind deposits, and beaches.

### 2.3. Tourism on the Canary Islands

The Canary Islands are currently a low-budget global tourism destination [61], this being a key as industry, it contributed most to the GDP (35.5%) and generated 40.4% of jobs [62]. In a territory that does not even reach 7500 km$^2$ and has a population of 2.2 million (in Gran Canaria, specifically, 870,595), the Canary Islands in 2019 received almost 13.1 million foreign tourists (National Statistics Institute, NSI) and approximately 2 million visitors from the rest of Spain. As shown in Figure 2, tourist arrivals grew until 2017, but then reported a decrease in 2018 and 2019. The islands that monopolize most of the tourist movement are Tenerife and Gran Canaria. In the case of the latter, tourist arrivals in 2019 amounted to 4,189,013. Most of them came from the Netherlands, Spain, United Kingdom, Germany, and the Nordic countries, as is shown in Table 1. All this in a context where the Canary Islands stand out for their high number of natural areas with some form of protection, specifically 146, which represent 40% of their territory.

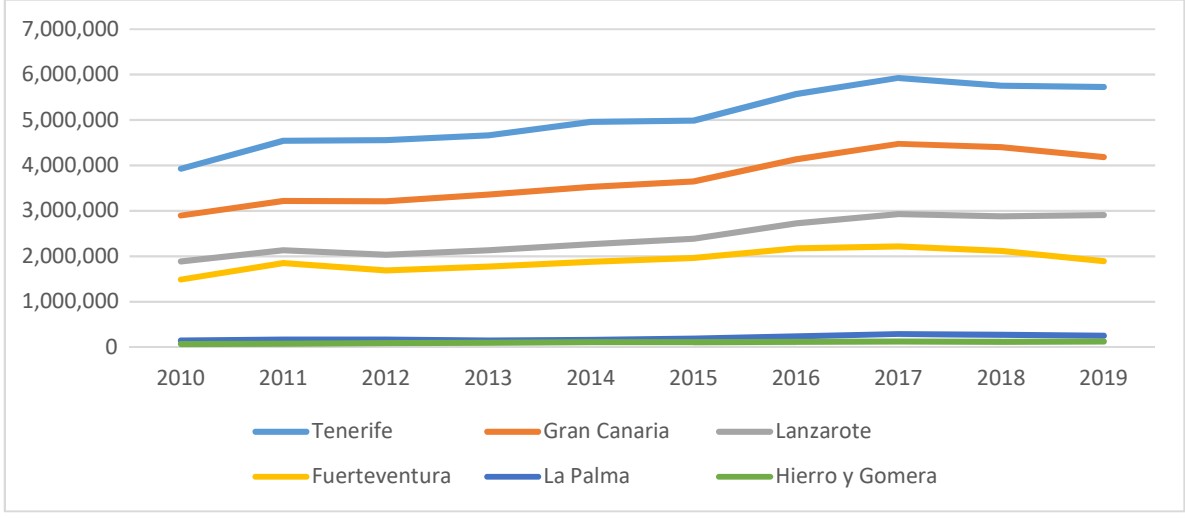

**Figure 2.** Tourist arrivals to the different islands (2010–2019). Source: Own elaboration with data from Turismo de Canarias (Canary Islands Tourism).

**Table 1.** Tourists by nationality (in thousands) arriving in Gran Canaria in 2019. Source: Own elaboration with data from Turismo de Canarias (Canary Islands Tourism).

| | |
|---|---|
| The Netherlands | 235,400 |
| Spain | 628,000 |
| United Kingdom | 759,400 |
| Germany | 852,900 |
| Nordic countries | 940,600 |
| Other countries | 772,713 |
| Total | 4,189,013 |

Analyzing the tourist arrivals to Gran Canaria during 2019, shown in Figure 2, Nordic, German and British people form the largest group, followed by the Spanish and Dutch.

## 2.4. Summer 2019 Fires

In summer 2019, serious fires destroyed an area of 71,486.80 hectares [63] and led to severe consequences for the environment. Throughout 2019, 10,883 fires were recorded, of which 7290 (66.98% of the total) were attempted, and 1 hectare or less burned. The remaining 3593 fires exceeded 1 hectare of burned area, and 14 affected more than 500 hectares, which places them in the category of large forest fires. Specifically, between 10 August and 26 September, the island of Gran Canaria suffered several forest fires that affected a significant area of the territory. Forty days of fires were divided into four areas, with two main and two secondary forest fires: Artenara, Valleseco, Los Cazadores, and Las Lagunetas. The main ones were in Artenara and Valleseco, which were later placed in the category of great forest fires. The two other forest fires of lesser impact, developed at the same time, were in Los Cazadores and Las Lagunetas.

The flames affected the municipalities of Vega de San Mateo, Tejeda, Valleseco, Artenara, Moya, Agaete, and Santa María de Guía, Gáldar. Around 10,000 people had to evacuate the areas, and more than 1000 human resources mobilized from different units, such as Civil Protection, the Fire Department of the Canary Islands, and the Military Emergency Unit (UME). During the extinction, there were 16 air means, 4 seaplanes, 1 forest plane, and 11 helicopters [63]. The affected tangible resources amounted to 91 properties, with economic losses valued at EUR 2.5 million [64].

The provided data by the Copernicus Project [65], as Figure 3 shows, illustrate the magnitude of the fire on the island of Gran Canaria during August, which burned a total area of 9636.40 hectares. Fires in Artenara and Valleseco accounted for 1137.60 and 8498.80 hectares, respectively. In global figures, according to data provided by MAPA [63], the fires in Gran Canaria represented 13.64% of the total area burned in the country during 2019. The area of destroyed trees accounted for 16.13% of the Spanish total, with the destruction of 40% of the trees in the Tamadaba National Park being particularly serious, as it represented 3000 hectares of high ecological value. The high temperatures, low air humidity, winds of more than 80 km/h, and difficult orography complicated the extinction of the fires.

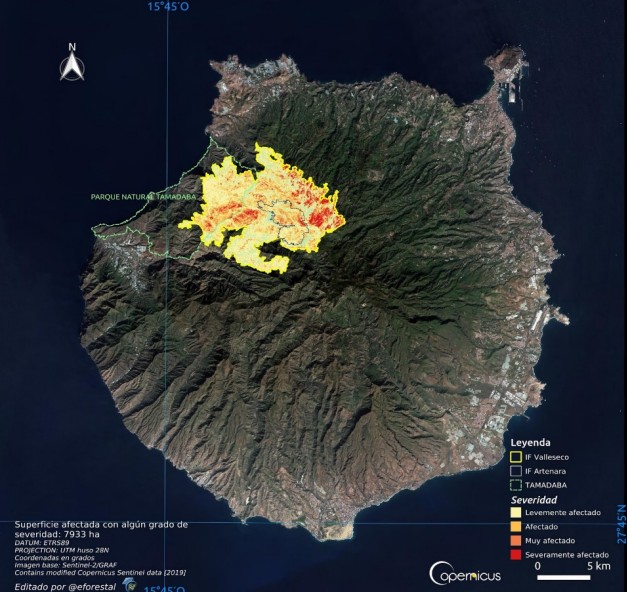

**Figure 3.** Fires in Gran Canaria and their severity, 25 August 2019. Source: Copernicus Service information [2018] for Copernicus Service Information. Edited by: @eforestal, member of Copernicus Academy (https://twitter.com/eforestal/status/1168965512564547584).

## 3. Materials and Methods

In this research, different tools were used to analyze the impact of Twitter on communication management during the aforementioned forest fires. In this case, the used tools were:

- Union metrics: This tool measures the reach of messages posted on Twitter by users, and the engagement that they generate among followers. It is a marketing tool made to help departments improve their knowledge of their competitors. In this case, it was useful in knowing the actual reach of the analyzed accounts.

- Twlets: This tool works as a Google Chrome widget and allows for the download of Twitter posts through a Python script. In addition, it measures and analyzes the sentiment through the VADER algorithm, a model developed by Hutto and Gilbert [65]: "a simple rule-based model for general sentiment analysis, and compare its effectiveness to eleven typical state-of-practice benchmarks including LIWC, ANEW, the General Inquirer, SentiWordNet, and machine-learning-oriented techniques relying on naive Bayes, maximum-entropy, and support-vector-machine (SVM) algorithms". VADER consists of qualitative and quantitative methods.

Following Gilbert and Hutto [65], due to the fact that the manual creation and validation of a comprehensive sentimental lexicon is labor-intensive and time-consuming, many automated means were explored to identify relevant sentimental characteristics in the text. Traditional state-of-the-art practices include machine-learning methods to "learn" the feeling properties of a text.

At this point, machine learning is relevant, but machine-learning approaches are not without drawbacks. In Gilbert and Hutto [65], there are four requirements. First, they require "training data that, like validated lexicons or feelings, are sometimes difficult to acquire"; second, they depend on the training set to represent as many characteristics as possible (which they often do not, especially in the case of short, scarce social-media text). The next requirement is that they are often more computationally expensive in terms of CPU processing, memory requirements, and training/classification time (which restricts the ability to judge the sentiment about the transmission of data). Lastly, they usually derive characteristics "behind the scenes" within a black box that are not interpretable by humans, so it is more difficult to generalize, modify, or expand. The acronym VADER stands for Valence Aware Dictionary for sEntiment Reasoning, and the algorithm is "a combination of qualitative and quantitative methods to produce and then empirically validate a gold-standard sentiment lexicon that is especially attuned to microblog-like contexts" [65]. It is shown on the Figure 4 as a flowchart

As a methodology, once the big data had been analyzed through the aforementioned tools, content analysis was performed. Authors Wimmer and Dominick [66] define this methodology as "a systematic procedure" designed to explore the content of archived information. Content analysis helps to make valid and stable inferences from context-related data. The definition offered by Kerlinger is the most standardized according to Wimmer and Dominick [66]: "Content analysis is a method of studying and analyzing communication in a systematic, objective, and quantitative way, with the purpose of measuring certain variables".

In this research, there is analysis of Twitter posts from various accounts related to natural-disaster management. The time limit was established as between 17 August, when the forest fire in Valleseco was declared, and 26 September 2019, when the extinction of this same fire was announced. In this case, the chosen accounts of the social media are:

- @avtorresp: verified user of Ángel Víctor Torres, president of the Canary Islands;
- @GranCanariaCab: verified official account of the governing body of the island of Gran Canaria;
- @ProteCivilLPA: official account of the Association of Civil Protection Volunteers of Las Palmas de Gran Canaria;
- @UMEGob: official profile of the Military Emergency Unit of the Armed Forces;
- @DgCanarias: official Twitter account of the Spanish Government delegation on the Canary Islands;
- @BomberosLPA: official account of the Fire and Rescue Service of Las Palmas de Gran Canaria;

- @012GobCan: Citizen Assistance Service of the Government of the Canary Islands; and
- @112Canarias: official profile of information of the Security and Emergency Department of the Canary Islands Government.

A total of 1739 posts were collected, 704 of which were retweets. Those unrelated to the fires were filtered out of the remaining 1005 tweets, which were 547, resulting in 458 analyzed posts. In order to carry out indepth content analysis, Table 2 was used.

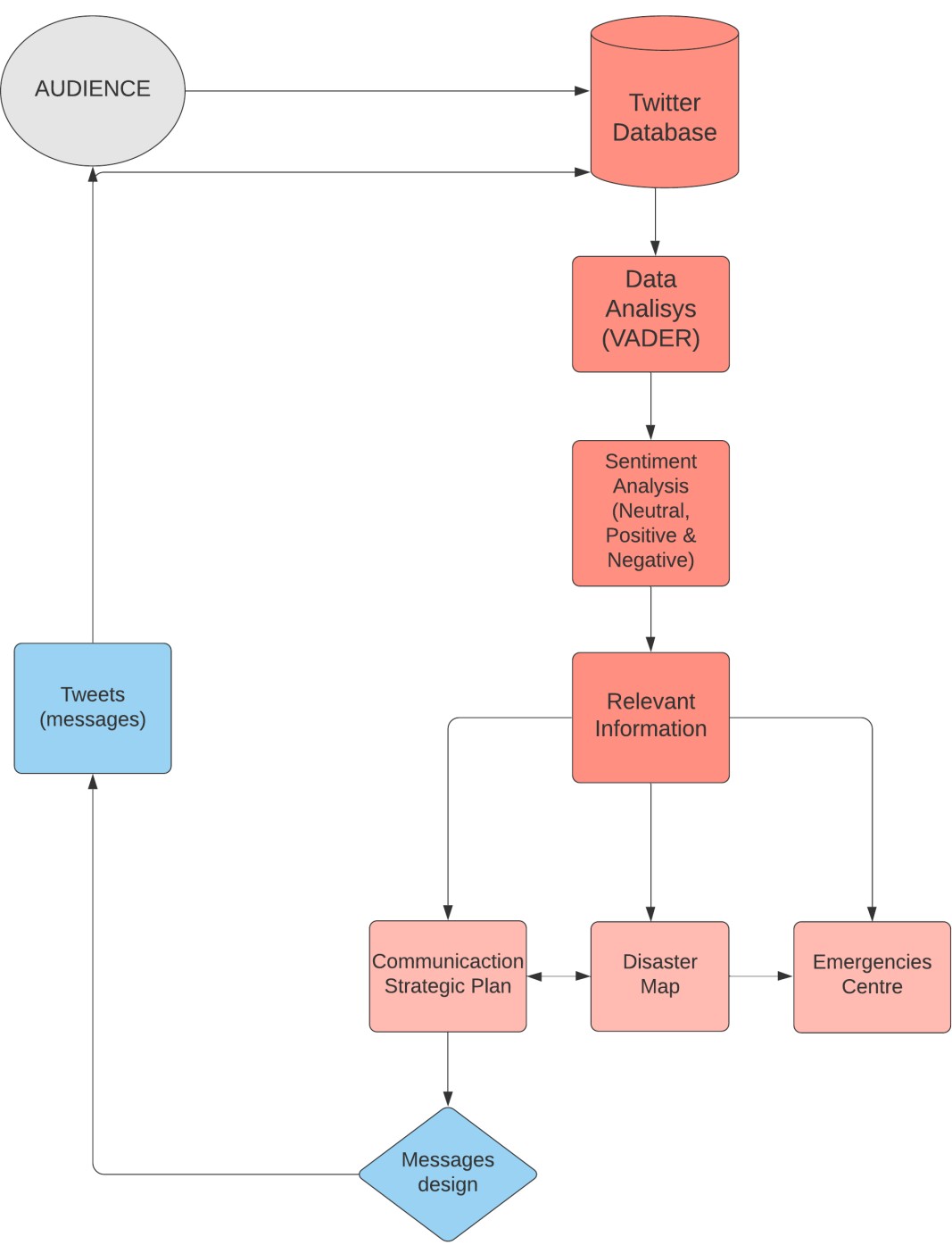

**Figure 4.** Flowchart. Source: own elaboration.

**Table 2.** Categories and subcategories of content analysis. Source: own elaboration, adapted from Cristòfol, F.J.; de -San-Eugenio-Vela, J., and Paniagua-Rojano [67].

| Categories | Subcategories |
|---|---|
| Date | Date of posting of tweets between 17 August and 26 September 2019. |
| Language | Spanish, English, other. |
| Content: type of message | • Advice: notices and advice to citizens.<br>• Info: general information about the situation of the fire.<br>• Specific for media: information addressed to the media, such as press conferences, calls, or statements to the press by politicians and technicians. |
| Qualifier: Sentiment conveyed through the message | • Concern: concern about the fires.<br>• Relief: messages of calm or relief about the fires.<br>• Gratitude: expressions of gratitude for citizen collaboration.<br>• Promoting the territory: messages related to the summer tourism campaign.<br>• Preventing future disasters: tips and actions to prevent other disasters.<br>• Request for help: call for citizen collaboration.<br>• Information for victims: tweets aimed at collecting information on victims and offering help to them.<br>• Public service information: posts that updated on the situation of the fires.<br>• Solidarity: messages of solidarity with the victims and affected people. |
| Attached | • Pic: tweets with pictures.<br>• Video: tweets with videos.<br>• No attachments. |

## 4. Results

The first measured element was the reach of the chosen accounts, as shown in Table 3. According to Union Metrics, they presented the following potential reach, where the most active account about the forest fires in Artenara and Valleseco had the most reach, this being the official profile of information of the Security and Emergency Department of the Government of the Canary Islands.

**Table 3.** Potential reach of Twitter accounts. Source: Union Metrics. Own elaboration.

| Account | Potential Reach | Related Posted Tweets |
|---|---|---|
| @avtorresp | 97.428 | 6 |
| @GranCanariasCab | 126.254 | 132 |
| @ProteCivilLPA | 18.889 | 14 |
| @UMEgob | 183.381 | 39 |
| @DgCanarias | 41.148 | 21 |
| @BomberosLPA | 236.532 | 38 |
| @012GobCan | 144.284 | 5 |
| @112canarias | 279.386 | 203 |
| | | Total: 458 |

Big-data analysis provided by Twlets through the implementation of a Python script regarding the general sentiment of the messages was at an average of −1.1%, assuming a markedly neutral tone in the posts. According to the VADER algorithm, the average sentiment was 2.3% negative messages, 96.1% neutral messages, and 1.6% of positive messages. On the other hand, tweets on average received 375.7 reactions, of which 259.3 were marked as favorite and 116.4 were retweeted, which is a somewhat evident result, as these are messages issued by official accounts. As for obtained results from content analysis (Figure 5), 93.67% of the messages were in Spanish, while only 29 tweets, 6.33% of the 458 tweets analyzed, were in a different language, specifically 3.93% in English and 2.4% in German.

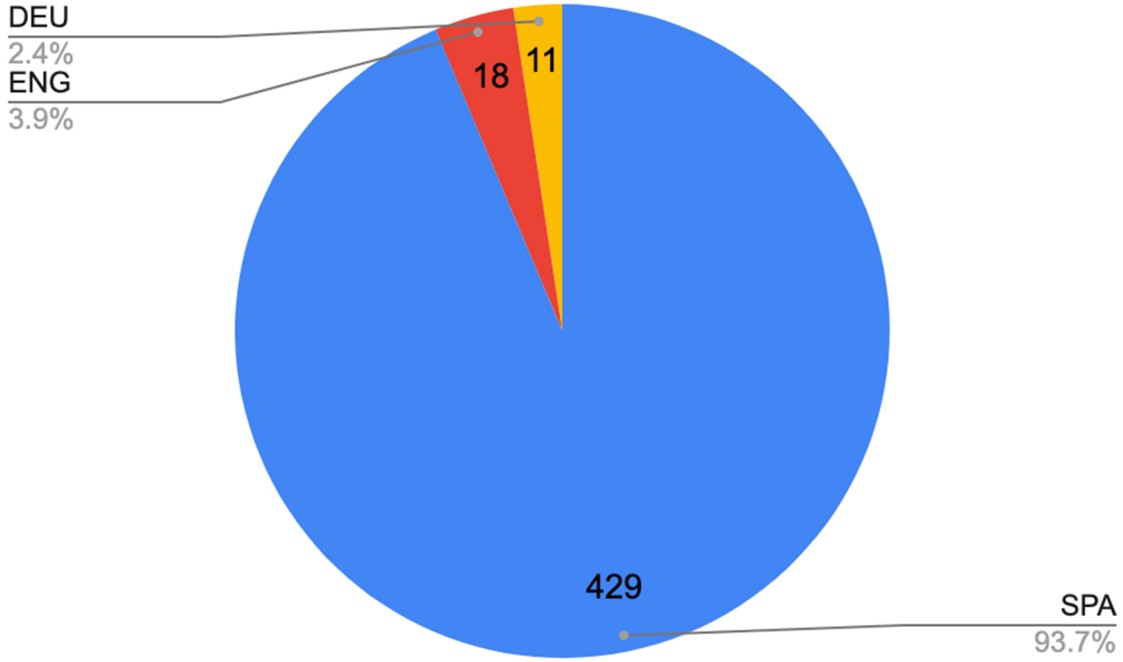

**Figure 5.** Tweets by language. Source: own elaboration.

In terms of the type of posted message (Figure 6), 65.7% (301) of the 458 tweets focused on the transmission of information about the fires, such as live updates to keep citizens aware of what is happening. On the other hand, authorities provided advice so that users could not perform their actions and stay away from danger areas or act responsibly, which, in this case, accounted for 20.7% of messages (95 tweets). Lastly, there was constant coverage in collaboration with the media and information addressed to the media, such as press conferences, calls, or press statements by politicians and technicians, accounting for 13.5% of the posted messages (62 tweets).

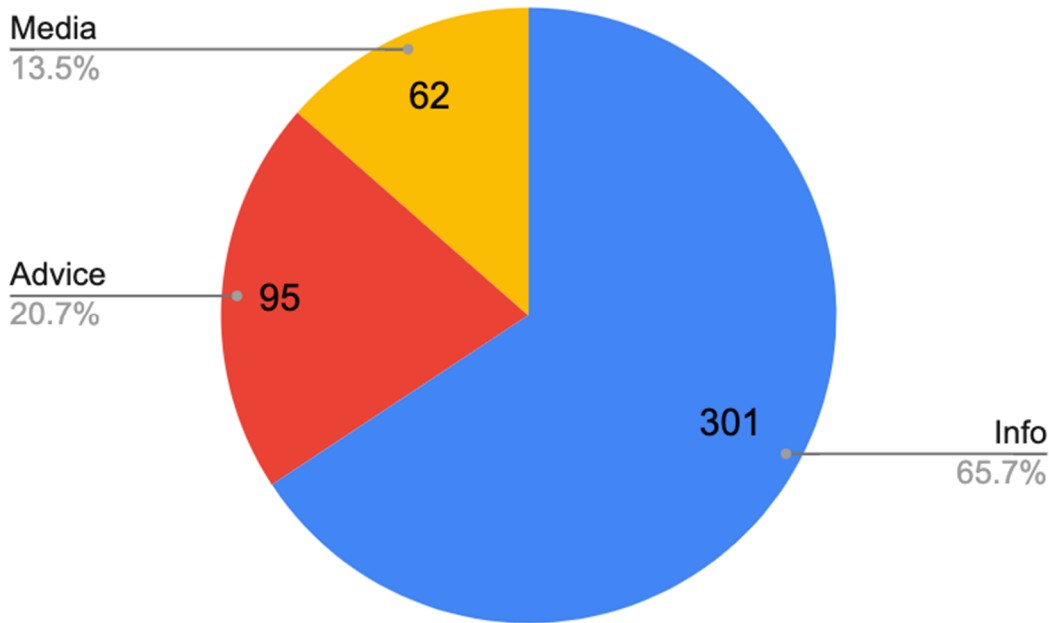

**Figure 6.** Type of message. Source: own elaboration.

With regard to the qualification of the messages, as in the case of the type of message (Figure 7), public service information predominated, with 72.5% of the total. This left a minimal presence of other concepts such as tourism promotion, which accounted for 7.4%, or messages explicitly linked to the prevention of natural disasters, which accounted for 4.6% of the total, with 21 posts out of 458.

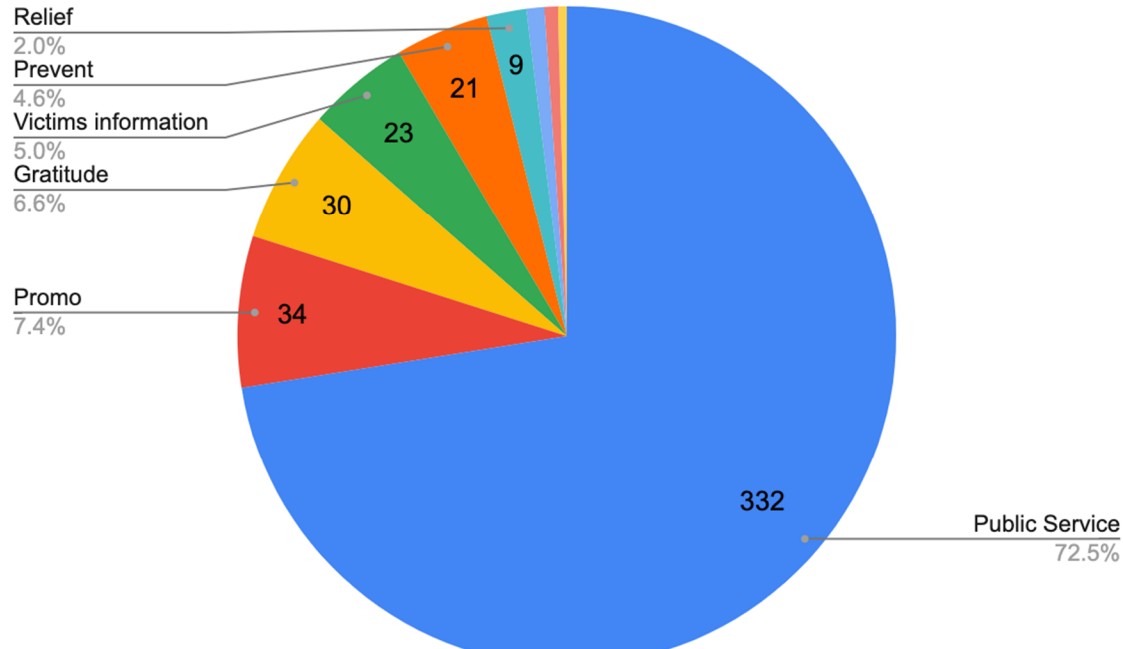

**Figure 7.** Tweets by qualifier. Source: own elaboration.

As for the presence of attached multimedia in the tweets (Figure 8), more than half (52.2% of those posted) did not include either a picture or video, 239 messages. Pictures appeared in 33.6% of the posts, and videos in only 14.2%.

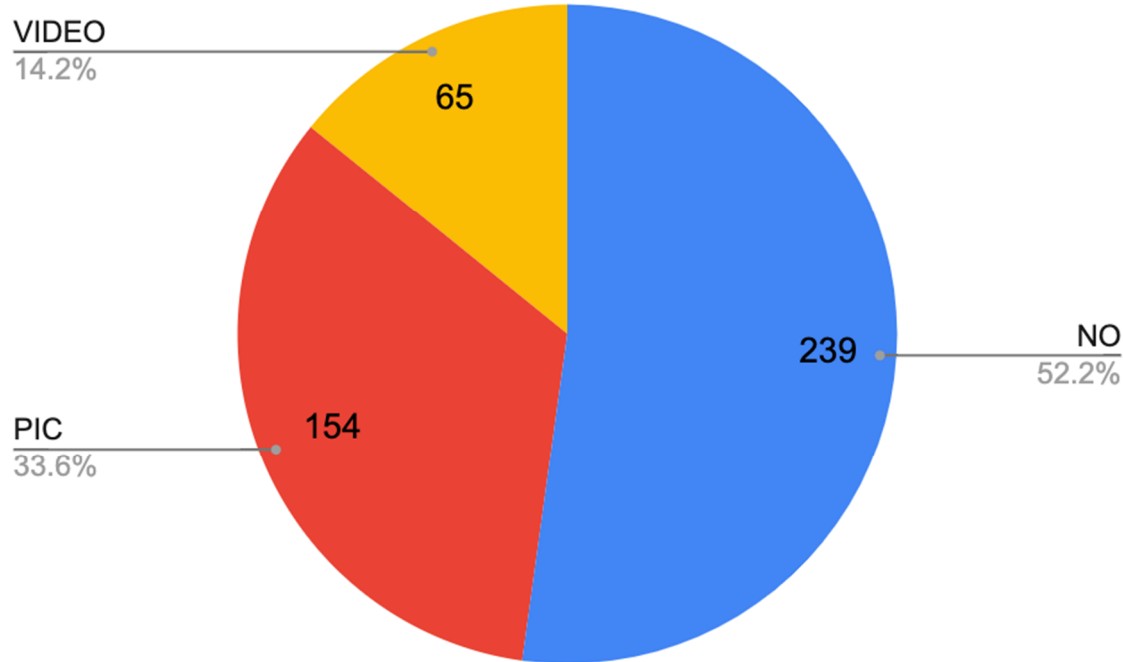

**Figure 8.** Tweets by attached. Source: Own elaboration.

Lastly, concerning the presence of words (Table 4), the hashtag #IFValleseco (IF for "forest fire" in Spanish) appeared in 40.2% of the posted messages by the studied accounts, while the hashtag #IFArtenara, only appeared in 5.9% of the tweets. The words "prevention" and "preventing" only appeared on three occasions.

**Table 4.** Tweets by words. Source: own elaboration.

|  | Total | % |
| --- | --- | --- |
| Fire | 75 | 16.4 |
| #IFValleseco | 184 | 40.2 |
| #IFArtenara | 27 | 5.9 |
| Prevention | 2 | 0.4 |
| Preventing | 1 | 0.2 |

## 5. Discussion: Open Innovation, Risk Management, and Communication Planning

On the basis of results, and regarding RQ1, the Canarian authorities used social media, specifically Twitter, as a channel of information to citizens, but did not post any relevant communication about prevention. In fact, at no time was work done on the prevention of natural disasters through the transmitted information via Twitter, focusing all work on massively informing the population through public service messages in order to live-update the development of forest fires. The use of Twitter as a tool for contacting the media is relevant for the live broadcasts of press conferences. As Pourebrahim et al. [45] states, the main purpose of social media is usually to allow for the sharing of texts, pictures, and videos as a communication tool during natural disasters. Following Geng et al. [30], there is a cognitive difference due to the use of the Internet, and we found difference in spatial and temporal perception.

According to Kim et al. [46], during the forest fires in Artenara and Valleseco, Twitter worked as an easier channel to exchange knowledge [55] since there was a proven significant amount of information transmission from public services, and one-time communications with those affected by the disaster in the Canary Islands. These results also show a clear likeness to the theory of Ukkusuri et al. [47]

regarding the approach of the Canary Islands authorities via Twitter to the management of emergencies derived from a natural disaster for service information. According to Dahal [41], social media should be taken as a tool in national crises due to natural disasters.

As Yang et al. [50] indicated, the existence of influential accounts, such as the analyzed official accounts, is a relevant element for managing emergency information with clear language and in a coherent manner. This need to function as a catalyst for information is precisely what makes these accounts present already filtered information to the citizens, since they can analyze big data.

Following Cachia et al. [55], in open innovation in the communication processes, there is a link between social media and innovative activities. However, the context in which social media can be used for open innovation in the funnel of innovation should be further investigated. This is to say, Canarian authorities did not use the whole potential of social networks regarding risk management and natural-disaster prevention. According to Howe [54], public administrations should use open innovation and even corporate R&D as elements to solve problems in the communication and prevention of natural disasters.

In this sense, and regarding natural disasters, the use of social networks, and especially, Twitter, should allow for public officials to collect knowledge produced by different involved stakeholders, as well as to use all these knowledge synergies in order to establish mid- and long-term prevention public policies. On the one hand, as Von Hippel states, cooperating within stakeholders "has been described as an important source of innovation" for organizations [68]. On the other hand, Cachia et al. [55] highlighted that "the sheer volume of user-generated content available on social networks allows for sophisticated environmental scanning through data mining". Mount and Garcia Martínez [56] concluded that "collective intelligence also helps reduce cognitive bias by allowing users to focus on processes, problems, and solutions that occur naturally".

Regarding the language of the messages, most of them were in Spanish, and communication from influential accounts in German and English was almost symbolic. This is striking since, according to the NSI, of the 870,595 inhabitants registered in Gran Canaria, 82,554 were foreigners, in addition to the 4,189,013 non-national tourists that the island received in 2019 according to the Canary Statistics Institute (see Figure 2 and Table 1). Therefore, communication regarding the prevention of future risks is not adapted to the reality of the inhabitants and visitors.

The posted messages directly focused on the transmission of updates on the forest fires in Artenara and Valleseco, so there was no element regarding communication for the possible prevention of future disasters. There was, however, an intention to minimize the effects of the natural disaster, since the citizens had information about movements made by different means of extinction, evacuations, and road closures. In short, there was no explicit intention of preventing future disasters through Twitter.

Regarding RQ2, the communication strategy of the Canarian authorities through Twitter was the use of evidence and danger to generate messages about the prevention of effects on citizenship and tourism, which it does in an unusual way. The aim of the messages was to inform social-media users in order to make them cautious about their actions and movements.

Lastly, social media make big data accessible to the average citizen by generating a collective intelligence that is very useful for disaster management [55]. Social media work as translators or catalysts of big data, which authorities manage in order to reach citizens. Thus, information transferred to databases can be interpreted to improve the early response and rationalization of the available means of help. The stored data on Twitter allow for the establishment of a cluster that can be useful for the location of disasters and their different degrees of impact on the territory or potential victims.

The use of social media and indepth analysis of the data contained there must be the subject of careful strategic planning, both in the public or governmental sphere and in the private sphere (citizens, companies, and private organizations) that can actively participate in the support and reconstruction of generated damage by natural disasters. In this way, the use of social media and especially Twitter during catastrophic events can streamline and energize the available means, and can mitigate the various generated damages. Open innovation processes can help public administration to design an

adequate strategic communication plan that is useful to build a sense of community and return those affected "to normal".

These open innovation processes depend on a cultural change in the organizations [57]. This requires a complex knowledge of their dynamics for the development within the organizations, especially in cases where we research public administrations.

The planning of communication through social media and the analysis of obtained data through the interaction of different stakeholders has a steering effect on the intervention process, generating less resistance or opposition from some social sectors since it forms an adequate consensus that encourages cooperation between the involved parties.

## 6. Conclusions

This research concludes that the Canarian authorities did not use messages to prevent future natural disasters, and used a digital-communication strategy only on the basis of information and live updates on the development of the forest fires in Artenara and Valleseco. Therefore, we highlight the need to deeply investigate how we can use social media, and especially Twitter, as tools to sensitize citizens to be proactive in natural-disaster prevention, that is to say, how public authorities can surpass reactive social-media management in order to strengthen their capacity to proactively use social media.

Furthermore, the use of social media by a public administration, and their effectiveness in communicating with citizens in a proactive way require a change in the organizational culture of this administration [57]. Companies and administrations must adapt and assume changes that arise due to the use of technologies. In this vein, public administrations can also manage open innovation processes to improve their strategic policy planning. Above all, considering that social media allow for different stakeholders to establish knowledge synergies in order to strengthen collective intelligence [55].

In this vein, we finish this article with some recommendations for the management of social media by authorities during the event of natural disasters as a result of what was studied here.

1.  The management of social media during an emergency must prioritize informative messages using data available to the authorities to anticipate the development of a disaster and warn citizens.
2.  On the basis of transparency, the temptation to post excessively should not exist in order to avoid unnecessary information, and there must be a distinction between technical and service information. Messages must be useful for the followers and reflect information and prevention.
3.  Authorities must convert the enormous amount of data available into clear and direct language mainly accompanied by audiovisual elements that make the information comprehensible to all audiences. That is, transforming big-data language into information that citizens understand.
4.  Through their messages, authorities can either generate prudence or transfer fear to the citizens. They must choose the strategy according to the moment, so that messages can be transmitted in accordance with it.
5.  It is essential to adapt communication in social media to the type of user, which is why it is necessary to have knowledge of the average follower. In this way, a user of social media is a content generator, which implies the ability of intervention and interaction with the sender by the authorities.
6.  Analysis of Twitter databases is a unique opportunity for early care centers of disasters to map crises in real time, from their origins to the subsequent analysis of the impact of catastrophic events.
7.  Public administrations must face the cultural change that implies the implantation of social networks as a basic tool of communication with their stakeholders, that is, civil authorities must be prepared to face open innovation processes.

**Author Contributions:** Conceptualization: G.Z.-A. and F.J.C.; data curation: F.J.C.; formal analysis: J.d.-S.-E.-V., F.J.C., G.Z.-A., and X.G.; methodology: F.J.C.; project administration: X.G.; resources: F.J.C. and G.Z.-A.; software: F.J.C.; validation: J.d.-S.-E.-V. and X.G.; writing—original draft, G.Z.-A. and F.J.C.; writing—review and editing, J.d.-S.-E.-V. and X.G. All authors have read and agreed to the published version of the manuscript.

**Funding:** No funding for this project.

**Conflicts of Interest:** The authors declare no conflict of interest.

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
