# Peer review of "Social-Media Analysis for Disaster Prevention: Forest Fire in Artenara and Valleseco, Canary Islands"

_2199-8531, doi:10.3390/joitmc6040169_

Round 1
Reviewer 1 Report
The overall manuscript deserves publication, but some minor changes are required.
1) Examples of typos:
Page 1-Line25: it should be "April 18th, 1906"
Please check the whole text for further minor typos.
2) Enrichment for literature review:
Page 2-Line 54-56:
when you mention "Social media play an increasingly important role in the organization of monitoring and coordination devices [28] as they did in the catastrophic floods in the states of Uttarakhand and Himachal Pradesh (India) [29].", I would suggest to add that:
The key role of social channels in monitoring activities may be enhanced further by the opportunities in terms of data visualization and learning, which are aimed at preventing and /or improving the overall management practices [A] [B].
- [A] Palmieri, R., Giglio, C. (2015). Informal learning in online social network environments: An evidence from an academic community on facebook. Turkish Online Journal of Educational Technology
- [B] Palmieri, R., Giglio, C. (2015). Using social network analysis for a comparison of informal learning in three Asian-American student conferences. Turkish Online Journal of Educational Technology
Author Response
Dear reviewer,
Thanks a lot for your comments. We have changed all the typos as you recommend. We also have included the references from Palmieri & Giglio you reccomend.
We appreciate your comments.
Reviewer 2 Report
Dear Authors,
The submitted manuscript titeled” Social Media Analysis for Disaster Prevention: Forest Fire in Artenara and Valleseco, Canary Islands” contains results, which might interest the international audience. The aims and background of investigations, as well as study area are properly described.
However, I have some suggestions to improve the manuscript:
- I suggest to unify subchapters 2.1 and 2.2, especially that they have the same title.
- The name of countries in Figure 3 should be written in English. The axes should be described.
- In my opinion the Figures and Tables should be self-explanatory. Therefore captions should be enlarged and All abbreviations should be explained.
- In the chapter „Conclusions” I suggest to resign from detailed data (lines 340-341). Moreover, in my opinion this chapter should be more concise. Perhaps some sections might be moved into „Discussion”.
- In my opinion the comparisons of role of social media in other disaster perception such as flooding, earthquake etc., especially in other geographic regions would be valuable and interesting.
Please, look into below listed publication which might be useful in preparation this section of discussion
- Geng et al. 2020. Spatial-temporal differences in disaster perception and response among new media users and the influence factors: a case study of the Shouguang Flood in Shandong Province. NATURAL HAZARDS DOI: 10.1007/s11069-020-04398-7
- Dahal et al. 2020. "It helped us, and it hurt us"The role of social media in shaping agency and action among youth in post-disaster Nepal. JOURNAL OF CONTINGENCIES AND CRISIS MANAGEMENT DOI: 10.1111/1468-5973.12329
Author Response
Dear Reviewer,
We have taken into account the changes proposed by you:
- We have unified the successive chapters
- We have modified the language of Figure 3. In fact, we have converted it into a table.
- We have modified the conclusion and discussion chapters following your advice.
- We have included other studies about the perception of other kind of disasters.
Finally, we have included the two references from Geng et al. and Dahal et al. that you proposed
Reviewer 3 Report
Social Media Analysis for Disaster Prevention: Forest Fire in Artenara and Valleseco, Canary Islands
Thank you for the opportunity to revise the paper titled “Social Media Analysis for Disaster Prevention: Forest Fire in Artenara and Valleseco, Canary Islands”.
In this paper the authors explore the use of Twitter during a particular event in Spain, a forest fire in Canary Islands. The authors, by studying the accounts of a number of politicians, reach the conclusion that authorities did not use social media as a preventive element, but almost exclusively as a live information channel.
The paper seems to be well written and covering a relevant topic. The implications can be of interest. Some suggestions follow:
- I would recommend you to include a final paragraph in the introduction describing the structure of the paper.
- Please note that sections 2.1 and 2.1 have the same title.
- Also please note that there are two sections 3.
- The analyses of Twitter is useful, but did you check other social media? Maybe Twitter was not much used because other tools were used?
- I would recommend you to include a section describing the limitations and especially the potential avenues for further research that open up from your study.
Good luck!
Author Response
Dear Reviewer,
Thank you for your valuable comments. We have taken into account the changes proposed by you:
- We have included a final paragraph in the introduction describing the structure of the paper.
- We have unified sections 2.1 and 2.2.
- We have modified the numbering of the sections.
- We have used Twitter because it is, among others, a "social sensor" that allows "immediate expression of feelings and opinions" of users.
- We have modified the conclusions and discussion chapters to adapt them to your advice
Round 2
Reviewer 3 Report
I have no further comments for the authors
Author Response
Thank you for all your comments and support.